# Multi-Biofluid Approaches for cftDNA and cftRNA Biomarker Detection: Advances in Early Cancer Detection and Monitoring

**DOI:** 10.3390/cimb47090738

**Published:** 2025-09-10

**Authors:** Douglas M. Ruden

**Affiliations:** Department of Obstetrics and Gynecology, C. S. Mott Center for Human Growth and Development, Institute of Environmental Health Sciences, Wayne State University, Detroit, MI 48201, USA; douglasr@wayne.edu; Tel.: +1-313-577-6688

**Keywords:** cell-free tumor DNA, cell-free tumor RNA, DNA-PAINT, nucleosome-bound DNA, liquid biopsy, cancer biomarkers, exosomes, proteomics

## Abstract

Cell-free tumor DNA (cftDNA) and cell-free tumor RNA (cftRNA) are emerging as powerful biomarkers for cancer detection, monitoring, and prognosis. These nucleic acids, released into the bloodstream by tumor cells, carry cancer-specific genetic and epigenetic alterations and can be detected non-invasively. Detection before clinical diagnosis offers a unique opportunity for earlier intervention yet requires longitudinal cohort studies to establish pre-diagnostic biomarker profiles. Current technologies enable sensitive quantification of cftDNA and cftRNA, with spike-in controls allowing for absolute quantification of single nucleosome-bound cftDNA, addressing a key limitation in liquid biopsy assays. Advances, such as DNA-PAINT, now permit single-molecule resolution detection of point mutations and methylation patterns characteristic of cancer, while new proteomics methods can identify the tissue of origin of exosome-derived nucleic acid. This review discusses the state-of-the-art detection strategies for cftDNA and cftRNA, highlights the gaps in longitudinal sampling, and outlines future research directions toward integrating multiomic liquid biopsy approaches for improved early diagnosis, monitoring, and relapse detection.

## 1. Introduction

Cancer remains one of the leading causes of morbidity and mortality worldwide, with outcomes heavily influenced by the stage at diagnosis [1]. Traditional diagnostic approaches—including tissue biopsies, imaging modalities, and serum protein markers—often fall short in detecting early-stage disease or predicting recurrence [2]. Liquid biopsies, which analyze components of tumor-derived material circulating in the bloodstream, offer a promising, minimally invasive alternative for cancer detection, prognosis, and therapeutic monitoring [3,4].

Cell-free tumor DNA (cftDNA), found mostly bound to nucleosomes [5,6], and cell-free tumor RNA (cftRNA), found mostly in extracellular vesicles [7], have emerged as promising non-invasive biomarkers for cancer detection, monitoring, and prognosis. These nucleic acids are released into the bloodstream from tumor cells through apoptosis, necrosis, or active secretion. Because they carry tumor-specific genetic and epigenetic alterations, cftDNA and cftRNA offer a unique molecular snapshot of the tumor’s current state, potentially enabling earlier diagnosis and better prediction of disease progression than traditional imaging or tissue biopsy approaches [8].

In cancer management, the ability to detect and quantify cftDNA and cftRNA before diagnosis offers the possibility of identifying malignant disease at a stage when intervention is most effective [9]. This requires large-scale longitudinal cohort studies in which samples from healthy individuals are collected and monitored over time to identify those who later develop cancer [9]. Such studies could reveal early molecular events in tumorigenesis and define baseline biomarker profiles for different populations [9].

Beyond diagnosis, cftDNA and cftRNA can provide valuable information for monitoring therapeutic response and for detecting relapse before clinical symptoms or imaging findings appear [10]. In this context, sensitive and specific detection methods are essential, with technological advances continuing to push the boundaries of what is possible in liquid biopsy assays [10]. Spike-in Controls [11], for example, now allow for the absolute quantification of single nucleosome-bound cftDNA in serum or plasma, addressing a key analytical challenge for liquid biopsy research [12].

While this review focuses on cell-free nucleic acids, it is important to acknowledge that circulating tumor cells (CTCs) represent another class of liquid biopsy biomarker [13]. CTCs are intact tumor cells shed from primary or metastatic sites into the bloodstream, and they can provide complementary information about tumor biology, including cellular morphology, protein expression, and functional assays [13]. However, the biology, isolation techniques, and clinical applications of CTCs differ significantly from cftDNA and cftRNA [13]. Given these distinctions, the detailed discussion of CTC-based methods is beyond the scope of this review, though future integrated approaches may combine both CTC and cell-free nucleic acid analyses for comprehensive cancer profiling.

The following sections will examine the current state of detection technologies for cftDNA and cftRNA, including their use in pre-diagnostic and relapse monitoring contexts, with an emphasis on advanced methods for quantifying single nucleosome-bound cftDNA and for detecting RNA species from extracellular vesicles.

## 2. Results

This review is based on a comprehensive literature search of peer-reviewed publications from PubMed, Scopus, and Google Scholar, using search terms including “cftDNA detection”, “cftRNA”, “liquid biopsy”, “DNA methylation”, “epigenetic kits”, “Epicypher,”, and “longitudinal cancer biomarkers”. The studies included spanned preclinical and clinical applications across a variety of cancer types, with an emphasis on methodologies that demonstrate high analytical performance and potential for translational use.

For technology-specific sections, we focused on the most used and emerging commercial platforms. For cftDNA, these included CUT&Tag [14], CUT&RUN [15], and Spike-in Control assays [11], bisulfite conversion kits (e.g., Zymo Research, Qiagen), droplet digital PCR (ddPCR) [16], BEAMing (Beads, Emulsion, Amplification, and Magnetics), and hybrid-capture next generation sequencing (NGS) methods [17]. For cftRNA, we evaluated protocols employing exosomal RNA extraction (e.g., exoRNeasy, Qiagen), RNA-seq with UMIs (Unique Molecular Identifiers) [18], and transcript-specific reverse transcription with multiplex amplification (reviewed in [19]).

We also included recent data from longitudinal studies, such as the Circulating Cell-free Genome Atlas (CCGA), sponsored by GRAIL, LLC (Menlo Park, CA, USA) [20], and the DETECT-A trial [21], which have provided key insights into the timing and predictive power of cell-free nucleic acids. An early version of the Graphic Abstract was made by ChatGPT5, but was heavily modified by the author. ChatGPT5 was also used to improve the quality and readability of the text.

### 2.1. Detection Technologies for cftDNA

This section reviews the current and emerging methods for identifying and characterizing circulating tumor DNA in patient blood samples. It highlights approaches ranging from advanced sequencing and digital PCR to novel single-molecule and epigenetic profiling technologies that improve sensitivity, specificity, and clinical utility in cancer diagnostics (Table 1).

#### 2.1.1. Quantifying Nucleosome-Bound cftDNA Using EpiCypher SNAP Spike-In Controls

Cell-free tumor DNA (cftDNA) is often fragmented to mono- and di-nucleosomal lengths, making it essential to precisely quantify and characterize nucleosome-bound DNA in plasma [12]. EpiCypher’s SNAP (Semi-synthetic Nucleosome Spike-in) controls represent a powerful advancement in this area [11]. These spike-ins consist of synthetic mononucleosomes with defined DNA and histone modifications that mimic endogenous chromatin particles [11]. When added to plasma-derived cftDNA workflows [35], SNAP controls serve as internal standards to normalize assay variability and to calibrate recovery efficiency, enabling absolute quantification of single nucleosome-bound DNA fragments (https://www.epicypher.com/products/spike-in-controls/ (accessed on 15 August 2025)) [11].

Unlike CUT&Tag [14] and CUT&RUN [15]—which are more suitable for mapping histone modifications on chromatin within intact cells—SNAP spike-ins are directly applicable to low-input biofluids, like plasma [11]. They allow researchers to assess not only total cftDNA yield but to infer nucleosome positioning and histone post-translational modification (PTM) patterns in circulating tumor-derived chromatin [11]. This is critical for detecting epigenetic changes linked to cancer type, stage, or progression.

#### 2.1.2. DNA Methylation Detection

DNA methylation profiling is one of the most robust and specific biomarkers for cancer detection. Cell-free methylated DNA immunoprecipitation sequencing (cfMeDIP-seq) [29,36], enzymatic methyl-seq (EM-seq) [37], and bisulfite conversion-based platforms (e.g., Zymo EZ DNA Methylation, Qiagen EpiTect) are widely used [38]. These methods can identify tissue-specific methylation patterns that are highly enriched in cancer, such as those in SEPT9 (colorectal cancer) [39], SHOX2 (lung cancer) [40,41], and GSTP1 (prostate cancer) [42,43]. Commercial kits are now capable of single-base resolution methylation detection with ultra-low input, making them suitable for liquid biopsies.

#### 2.1.3. Fragmentomics and Size-Selective Enrichment

The size and structure of cftDNA fragments offer an additional diagnostic layer. Tumor-derived cftDNA often exhibits distinct fragmentation patterns (~147–166 bp, corresponding to mononucleosomes), jagged ends, and preferred cleavage motifs [44]. Size-selection and fragmentomics approaches, including automated microfluidic electrophoresis (e.g., Agilent TapeStation) or targeted sequencing of preferred size bands, can enrich tumor-specific cftDNA and improve mutation or epigenetic signal detection [44].

#### 2.1.4. Mutation Detection: Digital PCR and NGS Panels

Mutation detection remains a cornerstone of cftDNA diagnostics. Droplet digital PCR (ddPCR) and BEAMing offer allele-specific sensitivity for known hotspot mutations, down to 0.01% variant allele frequency [16]. Broader mutation profiling is enabled by hybrid-capture and amplicon-based next-generation sequencing (NGS) platforms, such as Guardant360 [45,46] and FoundationOne Liquid CDx [32,47]. Incorporating unique molecular identifiers (UMIs) into NGS libraries has significantly reduced error rates, making it feasible to confidently detect ultra-rare variants in plasma DNA [48,49].

#### 2.1.5. DNA-PAINT in Cancer Detection

DNA-PAINT (DNA-based Point Accumulation for Imaging in Nanoscale Topography) is an emerging super-resolution microscopy method that offers unprecedented sensitivity and resolution in detecting cancer-associated DNA alterations [50]. By using short, fluorescently labeled DNA “imager” strands that transiently bind to complementary “docking” strands, DNA-PAINT achieves nanoscale resolution imaging without requiring complex optics [51]. DNA-PAINT is not yet a routine early detection tool but shows potential by offering single-molecule resolution, methylation-specific detection, and multiplexing. •High Sensitivity and Multiplexing:DNA-PAINT can detect DNA sequences at extremely low concentrations (femtomolar range), making it ideal for identifying rare cancer-associated mutations in liquid biopsies. It also supports multiplexing, allowing for the simultaneous detection of multiple oncogenic mutations, methylation sites, or genomic rearrangements in the same sample [51].•Point Mutation and Methylation Detection:Recent developments enable DNA-PAINT to identify single-nucleotide variants and methylation patterns characteristic of cancer DNA [50]. Methylated DNA often has a higher melting temperature than unmethylated DNA, and pattern recognition approaches like those used in the GRAIL platform can be applied to DNA-PAINT datasets for classification of cancer type and stage [50].•Single-Molecule Resolution for Amplified DNA:DNA-PAINT can visualize amplified DNA sequences, chromosomal rearrangements, and telomeric repeats with high spatial precision, providing both quantitative and structural insights into tumor DNA architecture [51].•Exosome and Tissue-of-Origin Profiling:When applied to exosome-derived nucleic acids, DNA-PAINT can be combined with advanced proteomics and proximity-barcoding methods that profile the protein cargo of exosomes to identify their tissue of origin [52]. This integration allows for a determination of where the cftDNA originated, adding an important layer of diagnostic specificity [52].•Advantages Over Other Methods:Compared to fluorescence in situ hybridization (FISH) or PCR-based assays, DNA-PAINT offers superior spatial resolution, lower background noise, and reduced need for amplification, minimizing the risk of introducing errors [51]. The method’s flexibility enables adaptation for DNA, RNA, or protein targets, making it a versatile addition to the cancer biomarker detection toolkit [51].

Together, these technologies provide a multifaceted toolkit for analyzing cftDNA—each contributing to mutation, epigenetic, or structural biomarker development. SNAP spike-in controls enhance quantitative rigor, particularly in early detection or treatment monitoring contexts where changes in nucleosome-bound DNA abundance may reflect subtle tumor dynamics [11].

### 2.2. Detection Technologies for cftRNA

Detecting and quantifying cell-free tumor RNA (cftRNA) remains technically challenging due to its inherent instability and low abundance in plasma. Nevertheless, several methods have emerged to improve RNA preservation, enrichment, and analysis—making cftRNA a promising biomarker for cancer detection, especially for gene expression and fusion transcript profiling (Table 2).

#### 2.2.1. RNA Stabilization and Isolation

Because cftRNA is highly susceptible to RNase degradation, immediate stabilization after blood draw is critical. Specialized blood collection tubes (e.g., PAXgene Blood ccfDNA Tubes, Streck RNA Complete BCT) stabilize RNA for up to several days at room temperature [64,65]. Following plasma separation, RNA is extracted using kits optimized for low-abundance RNA species (e.g., Qiagen miRNeasy Serum/Plasma Kit, Norgen cfRNA/cfDNA Purification Kit), which co-purify small RNA, mRNA, and lncRNA [64,65].

#### 2.2.2. Exosomal RNA Enrichment

A significant portion of circulating RNA is packaged within extracellular vesicles, such as exosomes. These lipid-bound vesicles protect RNA from degradation and often carry tumor-specific transcripts. Kits such as ExoRNeasy (Qiagen) [66] and ExoQuick (System Biosciences) [67,68] isolate exosomal RNA, which can then be profiled using RNA-Seq, RT-qPCR, or digital PCR. Exosomal RNA has been shown to carry oncogenic transcripts, like EGFRvIII in glioblastoma [69,70], and fusion RNAs, such as EML4-ALK in lung cancer [71].

#### 2.2.3. Reverse Transcription and Amplification Technologies

Low-input RNA samples require highly efficient reverse transcription and cDNA amplification. The use of random primers, target-specific primers, or UMIs helps preserve transcript complexity and allows for quantitative tracking of individual RNA molecules. Platforms such as Takara SMARTer kits and Qiagen’s QIAseq miRNA library kits are optimized for ultra-low input and small RNA profiling.

#### 2.2.4. RNA-Seq and Digital PCR Applications

RNA-Seq is increasingly used to analyze cftRNA, particularly when targeting known cancer-associated gene panels. However, full transcriptome sequencing is often impractical due to low RNA yield [72]. As an alternative, targeted panels (e.g., AmpliSeq Transcriptome, Oncomine Focus RNA) enable deep coverage of oncogenic transcripts and fusions with high sensitivity [73]. Digital droplet PCR (ddPCR) offers a lower-cost and highly quantitative alternative, especially for recurrent fusion transcripts (e.g., BCR-ABL [74], TMPRSS2-ERG [75]) or splice variants [76].

#### 2.2.5. Circular and Non-Coding RNAs

Emerging research has indicated that circular RNA (circRNA) and long non-coding RNA (lncRNA) are stable and informative components of cftRNA [77]. These species are often overexpressed in cancer and resistant to exonuclease degradation (reviewed in [78]). Detection platforms are being developed to capture and sequence circRNAs and lncRNAs from plasma with high specificity [77].

Together, these RNA detection strategies enable increasingly sensitive, specific, and reproducible profiling of tumor-derived transcripts from peripheral blood. While still limited by stability and technical complexity compared to cftDNA, cftRNA offers dynamic expression-based biomarkers that may provide additional dimensions for cancer detection and prognosis.

### 2.3. cftDNA and cftRNA from Non-Blood Body Fluids

While blood plasma remains the most widely used and validated source for cftDNA and cftRNA analysis, emerging research has highlighted the potential of other body fluids as complementary or alternative sources for cancer detection, prognosis, and monitoring. These “non-blood liquid biopsies” can, in some cases, offer higher sensitivity for certain malignancies, particularly when tumors are anatomically close to the sampled fluid or when blood-based cftDNA and cftRNA are scarce (Figure 1) [79].

#### 2.3.1. cftDNA and cftRNA from Cerebrospinal Fluid (CSF) for CNS Tumor Detection and Monitoring

Cerebrospinal fluid (CSF) is increasingly recognized as a uniquely informative biofluid for the analysis of cell-free tumor DNA (cftDNA) in patients with primary and metastatic tumors of the central nervous system (CNS) [80]. The blood–brain barrier severely limits the release of tumor-derived nucleic acids into peripheral circulation, meaning that plasma-based cftDNA assays often yield false negatives or very low variant allele fractions in CNS malignancies [80]. By contrast, CSF—obtained via lumbar puncture or, less commonly, ventricular access—lies in direct contact with CNS tumors, including gliomas, medulloblastomas, and brain metastases from systemic cancers, such as lung, breast, and melanoma [80]. This anatomical proximity results in higher cftDNA concentrations, reduced dilution by non-tumor cfDNA, and lower interference from hematopoietic background DNA, thereby enabling more accurate mutation profiling [80].

The molecular analysis of CSF-derived cftDNA has demonstrated clinical utility in multiple domains, including the following: (1) improving diagnostic sensitivity when surgical biopsy is not feasible [81]; (2) enabling molecular subtyping and detection of actionable driver alterations, such as *IDH1/2* [82], *TERT* promoter [83], or *H3K27M* mutations in gliomas [84]; (3) guiding targeted therapy selection in metastatic CNS disease (e.g., *EGFR* T790M in lung adenocarcinoma brain metastases) [85]; (4) tracking minimal residual disease (MRD) and early recurrence before radiographic progression [84]; (5) monitoring the emergence of treatment resistance [86]. Beyond DNA point mutations, copy number changes, structural rearrangements, and methylation signatures in CSF, cftDNA, and cftRNA offer additional diagnostic and prognostic value.

Despite its advantages, CSF sampling for any disease carries the following limitations: lumbar puncture is moderately invasive and unsuitable for some patients [87]; cftDNA yields can vary with tumor location, size, and degree of leptomeningeal involvement [88]; standardized pre-analytical protocols (e.g., cell-free DNA stabilization, storage, and extraction methods) remain lacking for any bodily fluid [89]. Furthermore, while targeted NGS and digital PCR have shown high analytical sensitivity, ultra-low input requirements, assay harmonization, and cross-center reproducibility are still ongoing challenges [90]. Moreover, prospective, multi-institutional trials are needed to fully establish the clinical validity and utility of CSF-derived cftDNA as a frontline or complementary liquid biopsy modality for CNS malignancies [91].

#### 2.3.2. cftDNA and cftRNA from Urine for Bladder, Kidney, and Prostate Cancer

Urinary cftDNA offers a compelling, entirely non-invasive source for tumor-derived genetic material, particularly for cancers of the urinary tract, such as bladder [92], kidney [93], and prostate cancer [94], but also for distant malignancies that shed DNA into the circulation and subsequently into urine via glomerular filtration [95]. The key advantage of urine sampling is the ease and frequency with which it can be obtained, facilitating serial monitoring without the logistical or psychological barriers associated with phlebotomy. Tumor-derived DNA in urine exists both as cell-free fragments from plasma ultrafiltration (transrenal DNA) [95] and as locally shed DNA from exfoliated cancer cells within the urinary tract [92].

Analytical sensitivity can be challenged by lower DNA concentrations compared to plasma and the presence of high-molecular-weight genomic DNA from normal urothelial cells, which can dilute tumor-specific signals [96]. Pre-analytical variables, such as diurnal variation, sample volume, and the need for immediate stabilization of nucleic acids, are critical to preserve fragment integrity and methylation patterns [96]. Advanced enrichment strategies, including size selection to capture the shorter, tumor-derived fragments, and targeted methylation profiling, have improved detection rates for early disease and minimal residual disease (MRD) [96]. With standardized workflows, urinary cftDNA could complement plasma-based assays or serve as a frontline tool for cancers with direct urinary tract involvement.

#### 2.3.3. cftDNA and cftRNA from Saliva for Cancers of the Aerodigestive Tract

Saliva represents an underutilized but highly accessible medium for cftDNA analysis, particularly valuable for head and neck squamous cell carcinoma (HNSCC), oral cavity cancers, and other tumors within the aerodigestive tract that shed DNA locally into oral fluids [97]. Saliva-based cftDNA sampling is non-invasive, painless, and suitable for frequent home-based collection, offering a unique opportunity for population-level cancer screening and monitoring [98]. The tumor-derived DNA within saliva may be derived from local tumor shedding, gingival crevicular fluid, or transudation from plasma, creating a heterogeneous mixture of DNA fragment sources [98].

Challenges include enzymatic degradation by oral nucleases, contamination from oral microbiota, and variability in saliva production rates influenced by hydration, circadian rhythms, or glandular pathology [97]. Optimized collection devices with nucleic acid stabilizers and rapid downstream processing are crucial to preserving fragment size profiles and epigenetic signatures [97]. Molecular assays such as targeted NGS, droplet digital PCR, and methylation-specific PCR have shown high sensitivity for detecting tumor mutations and HPV DNA in HNSCC patients [99]. Saliva-based cftDNA, especially when paired with plasma analysis, could expand the reach of precision oncology into community and remote settings [100].

#### 2.3.4. cftDNA and cftRNA from Pleural Fluid for Malignant Cancers

Pleural effusions are often rich in tumor-derived DNA when caused by malignant processes, such as metastatic lung, breast, or ovarian cancers, or malignant mesothelioma [101]. Because pleural fluid directly bathes tumor deposits in the pleural space, cftDNA from this compartment may display higher variant allele fractions (VAFs) and greater tumor heterogeneity than concurrent plasma samples [102]. This makes pleural fluid a valuable source for genomic profiling when a tissue biopsy is not feasible or yields insufficient material [102]. In addition to somatic mutation detection, pleural fluid cftDNA preserves fragmentomic and methylomic signatures that may reflect the tumor’s local microenvironment [103].

However, the utility of pleural fluid is limited by its invasive collection method (thoracentesis), potential sampling bias due to loculated effusions [104], and variability in cftDNA yield depending on fluid cellularity and turnover rates [101]. Standardization of cell-free supernatant processing—separating cftDNA from cellular gDNA—is essential to prevent signal dilution [101]. Given its high tumor content, pleural fluid cftDNA shows particular promise for guiding targeted therapy selection, resistance mutation tracking, and real-time disease monitoring for pleural malignancies [79].

#### 2.3.5. Advantages of Non-Blood Fluids

Non-blood fluids offer several advantages:•Higher local cftDNA concentration when sampling near the tumor site (e.g., CSF for CNS tumors, urine for bladder cancer) [79].•Utility for inaccessible tumors, where tissue biopsy is high-risk or impractical [79].•Enhanced tumor heterogeneity profiling, as sampling from multiple fluids may capture distinct subclonal populations [101].•Facilitation of serial monitoring, particularly with easily collectible fluids, such as urine or saliva [97].

However, these approaches face technical and clinical challenges, including variability in cfDNA yield, the influence of local inflammation or infection on background DNA levels, and the need for standardized pre-analytical and analytical workflows [105,106]. While still in the translational phase for many cancer types, multi-fluid cfDNA and cfRNA analysis holds promise for expanding the reach and resolution of liquid biopsy-based oncology [105,106].

### 2.4. Clinical Trials and Longitudinal Studies of cftDNA Detection Methods

The clinical translation of cftDNA technologies is rapidly advancing, with multiple high-profile trials now underway to test their real-world performance in early detection, minimal residual disease (MRD) monitoring, and longitudinal surveillance. Below we highlight several landmark trials that collectively demonstrate the clinical and technological breadth of cftDNA research (Table 3).

#### 2.4.1. GRAIL CCGA (Circulating Cell-Free Genome Atlas, NCT02889978)

The GRAIL Circulating Cell-free Genome Atlas (CCGA) study provided one of the largest datasets on cftDNA methylation analysis across multiple cancer types. Reported sensitivities ranged from 39% in stage I to over 90% in stage IV, with a specificity close to 99% [107]. These findings highlight the stage-dependency of cftDNA detection: while highly reliable for advanced cancers, the main limitation remains the low sensitivity for the early-stage disease where clinical need is the greatest. Nonetheless, the tissue-of-origin predictions were highly accurate, providing a crucial framework for clinical application [107,110].

#### 2.4.2. NHS-Galleri Trial (ISRCTN91431511)

The NHS-Galleri trial (ISRCTN91431511) is the first large, population-based prospective screening study evaluating cftDNA-based methylation classifiers in over 140,000 asymptomatic individuals. Interim results demonstrated a positive predictive value (PPV) of ~43% with a specificity of 99.5% [108,111]. While this PPV might appear modest, in the context of population screening it represents a significant advance compared to conventional screening approaches, particularly given the breadth of cancer types detected. Ongoing results will determine whether the Galleri test reduces late-stage diagnoses, which remains the primary endpoint of clinical utility [108,111].

#### 2.4.3. Natera Signatera MRD Program (Multiple Trials)

Natera’s Signatera™ assay represents a tumor-informed, personalized cftDNA sequencing platform used primarily in minimal residual disease (MRD) detection. Across colorectal [112,113], breast [114,115,116,117], and lung [110,118] cancer cohorts, the Signatera test has demonstrated PPVs exceeding 95% for recurrence and a median lead time of 8–12 months compared with radiographic imaging. This illustrates the transformative role of cftDNA in surveillance and treatment stratification, particularly in guiding adjuvant therapy decisions where imaging and pathology provide limited resolution.

#### 2.4.4. Guardant Health ECLIPSE/LUNAR-2 (Shield, NCT04136002)

The Guardant ECLIPSE (LUNAR-2/Shield) trial focuses on the early detection of colorectal cancer in average-risk populations. Preliminary results have shown a sensitivity of ~83% and specificity of ~90%, with a PPV of ~49% in the screening setting [109]. These results compare favorably with traditional stool-based screening methods while offering broader insights into cancer biology through fragmentomics and methylation signatures. However, the ~10% false positive rate emphasizes the need for confirmatory diagnostic colonoscopy, underscoring a balance between sensitivity and specificity in real-world screening programs.

#### 2.4.5. Comparisons of cftDNA Clinical Trials

Taken together, these clinical trials have illustrated the central trade-off that defines current cftDNA technologies. Specificity and PPV are consistently high across platforms, approaching or exceeding 95% in tumor-informed MRD assays, such as Signatera, and ~99% in population studies, like CCGA and NHS-Galleri. However, sensitivity remains strongly stage-dependent. Early-stage cancers, where detection has the greatest clinical impact, continue to pose challenges, with sensitivities in the 30–40% range for stage I compared to >90% for the advanced disease.

Tumor-informed approaches (e.g., Signatera) achieve excellent PPVs for recurrence monitoring but are less applicable to primary screening, whereas tumor-naïve methods (e.g., Galleri, Guardant Shield) broaden applicability but face higher false-positive rates in low-prevalence populations. These findings emphasize that no single assay currently achieves optimal performance across all use cases. Instead, clinical deployment must be tailored—tumor-informed methods for surveillance and MRD, tumor-naïve methylation or other classifiers for early detection—while ongoing advances in assay sensitivity, biofluid selection, and multiomic integration will be critical to bridging these gaps.

### 2.5. Retrospective Pre-Diagnosis Studies of cftDNA (From Banked Samples)

A small but growing body of the literature shows that tumor-derived signals can be detected in archived, pre-diagnosis plasma samples, supporting the biological plausibility of earlier detection with cftDNA (Table 4). Three major exemplars span multi-cancer discovery cohorts and population-based screening analogs. The PanSeer (Taizhou longitudinal cohort) case-control study demonstrated methylation-based detection up to four years before clinical diagnosis across several cancer types [119]. The HUNT nested case-control study mimicked a screening setting for colorectal cancer (CRC), detecting methylated cftDNA markers up to two years pre-diagnosis with modest sensitivity at useful specificity [120]. Most recently, the ARIC study leveraged serial banked samples and duplex-sequencing-based MCED methods, showing that mutations present months before diagnosis were often traceable >3 years earlier at ~10–80× lower VAFs; no controls were positive in the small-matched set, yielding a dataset-specific PPV of 100% that should not be generalized [121]. Together, these studies illustrate feasibility, underscore sensitivity limits at very low tumor fractions, and motivate large prospective, longitudinal cohorts with standardized pre-analytics and orthogonal confirmation.

## 3. Discussion

The field of circulating nucleic acid biomarkers has evolved rapidly over the past decade, driven by technological advances that increase sensitivity, specificity, and the amount of information retrievable from small plasma volumes. Among the most promising advances for cftDNA detection are SNAP Spike-in Controls [77], which enable the quantitative measurement of single nucleosome-bound DNA in plasma. Unlike conventional ChIP-seq-based methods, these controls are optimized for fragmented DNA typical of plasma samples, providing a rigorous reference for methylation and histone modification studies [77]. Their use addresses a persistent challenge in plasma cfDNA research: distinguishing true biological variation from technical noise.

Spike-in controls, such as SNAP-based nucleosome quantification standards [77], represent a significant refinement in cftDNA quantification by reducing assay-based misestimation and improving reproducibility across platforms. While they do not directly overcome the challenge of low analyte concentrations, they provide a necessary step toward standardization, which is essential for clinical translation.

DNA-PAINT offers an entirely different advantage—single-molecule resolution combined with multiplexing capability [51]. Its potential in liquid biopsy is twofold: direct detection of point mutations and methylation profiling from individual cftDNA molecules, and the possibility of identifying DNA fragment patterns indicative of tissue-of-origin [50]. The latter is particularly promising when integrated with emerging proteomic approaches to analyze exosomal protein cargo [52], which can reveal the tissue type from which the DNA or RNA originated. Such an integrative analysis could transform both early detection and relapse monitoring.

Despite these advances, significant challenges remain, as detailed in the next section (Figure 2). Biological variability, tumor heterogeneity, and technical inconsistencies can complicate interpretation. The requirement for ultra-sensitive detection also increases susceptibility to contamination and background signals. Furthermore, while current methods excel in retrospective analyses, their real-world performance in prospective, longitudinal cohorts is less well-established.

## 4. Limitations and Future Directions

Future research on cftDNA and cftRNA biomarkers should prioritize multiomic integration. Combining single-molecule DNA-PAINT for genetic and epigenetic profiling with proteomic tissue-of-origin mapping can create comprehensive diagnostic signatures [50]. Longitudinal cohort studies should be designed to collect pre-diagnostic and post-diagnostic plasma samples to identify early molecular changes that precede clinical symptoms. Although technologies such as DNA-PAINT have yet to overcome the low input and false-positive challenges inherent to early-stage cftDNA detection, their ability to interrogate single molecules and epigenetic modifications suggests they may ultimately complement enrichment and multiomic strategies in future clinical assays.

However, several limitations remain before these approaches can be fully implemented in clinical practice (Figure 2). One major challenge is the low concentration of cftDNA and cftRNA in the bloodstream during early cancer stages or after effective treatment, necessitating highly sensitive detection platforms. Additionally, there is currently a lack of standardization in sample collection, storage, assay procedures, and data analysis workflows, making cross-study comparisons difficult and slowing regulatory approval.

Cost and accessibility also remain significant barriers; many advanced detection methods require specialized equipment and expertise, limiting their adoption in routine diagnostics. Furthermore, the risk of false positives and false negatives persists due to the difficulty of distinguishing tumor-derived nucleic acids from background material shed by healthy cells or arising from benign physiological conditions.

Addressing these limitations will require coordinated efforts to establish standardized operating procedures and quality control measures. Automation of workflows, including SNAP Spike-in-based nucleosome quantification, could enhance scalability and reproducibility. For cftRNA specifically, the development of more robust RNA preservation and isolation methods will be essential to maintain integrity and to maximize diagnostic yield.

Another priority is expanding bioinformatic frameworks capable of integrating large, multi-modal datasets from DNA, RNA, and proteomics. Machine learning could be leveraged to detect subtle, clinically relevant molecular patterns predictive of disease progression or relapse. Finally, regulatory frameworks must evolve alongside technological advances to ensure that new assays are validated rigorously but adopted rapidly enough to benefit patient care.

## 5. Conclusions

The integration of circulating tumor DNA and RNA analyses across both blood and non-blood biofluids holds significant promise for advancing early cancer detection, monitoring therapeutic response, and guiding personalized treatment strategies. While current challenges—such as assay standardization, sensitivity in low-tumor-burden settings, and validation in large, diverse cohorts—must be addressed, ongoing technological innovations and multiomics approaches are rapidly enhancing the clinical utility of these biomarkers. Ultimately, the convergence of improved detection platforms, robust bioinformatics pipelines, and expanded biofluid sampling will enable liquid biopsy technologies to transition from research tools to routine components of precision oncology.

## Figures and Tables

**Figure 1 cimb-47-00738-f001:**
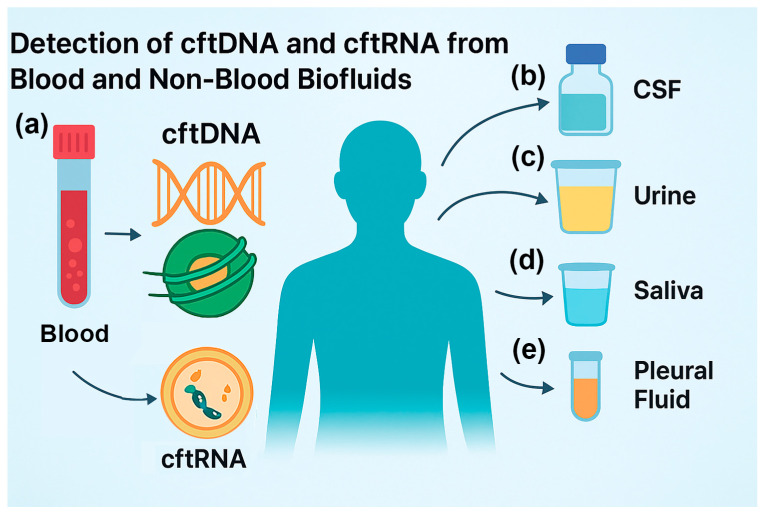
Detection of cftDNA and cftRNA from blood and non-blood biofluids: (a) isolation of cftDNA and cftRNA from blood and serum, such as for multiple cancer detection methods; (b) isolation from cerebral spinal fluid (CSF), such as from brain cancer patients; (c) isolation from urine, such as from bladder cancer patients; (d) isolation from saliva, such as from throat cancer patients; (e) isolation from pleural fluid, such as from lung cancer patients.

**Figure 2 cimb-47-00738-f002:**
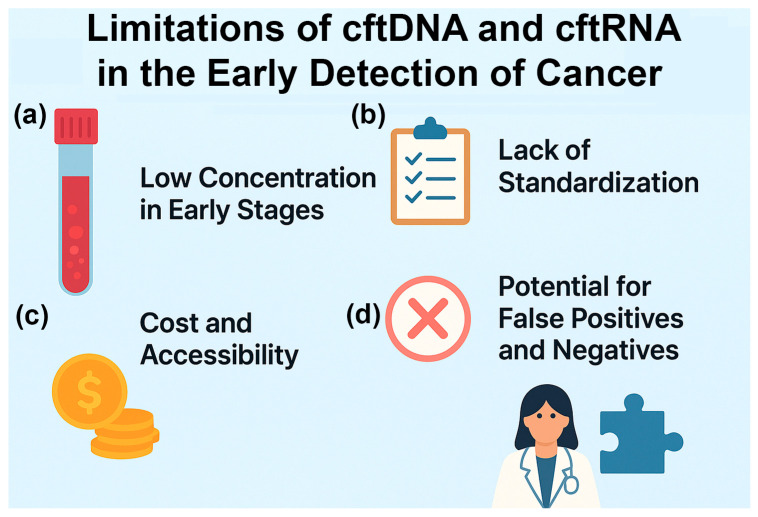
Limitations of cftDNA and cftRNA in the early detection of cancer: (a) Low concentrations in early stages. In the early stages, when the tumor is small, there will be low to undetectable levels of cftDNA and cftRNA in the biofluids. (b) Lack of standardization. Biofluids, cfDNA, EVs, cfRNA, and detection methods vary according to the laboratory. (c) Cost and accessibility. The kits and NGS assays are typically expensive and usually not covered by insurance. (d) Potential for false positives and negatives. The assays have a high rate of false positives and negatives, making it difficult for doctors to prescribe treatments based on the results.

**Table 1 cimb-47-00738-t001:** cftDNA detection methods, short description, and references.

Method	Short Description	References
CAPP-Seq	Cancer Personalized Profiling by deep Sequencing; targeted hybrid-capture NGS selector approach designed for ultrasensitive, broad patient-coverage quantification of ctDNA (mutation, indel, rearrangement, CNV).	[22]
iDES	Integrated digital error suppression—molecular barcodes (UMIs) + in silico background filtering to reduce NGS errors and lower limit of detection.	[23]
BEAMing	Beads, Emulsion, Amplification, Magnetics; emulsion PCR + bead capture and flow cytometry readout for ultra-sensitive detection of known hotspot mutations (digital PCR class).	[24]
ddPCR	Droplet digital PCR; partitioned PCR (droplets) for absolute quantification of known single nucleotide variants (SNVs) and small insertion–deletions (indels); highly sensitive for low variant allele frequencies (VAFs) and orthogonal validation.	[25,26]
sWGS	Shallow whole-genome sequencing; low-coverage WGS to detect genome-wide copy number alterations and tumor fraction estimation from cfDNA.	[27]
Fragmentomics (DELFI)	Genome-wide cfDNA fragmentation profiling (fragment size, end-motifs, nucleosome signals) used with machine learning (ML) to detect and localize cancer.	[27]
cfMeDIP-seq	DNA methylome profiling; immunoprecipitation-based enrichment for methylated cfDNA enabling low-input methylome profiling and tissue-specific cancer signals (bisulfite-free options).	[28,29]
cfMeDIP-spike	Synthetic spike-in DNA controls (variable length/GC/CpG) for normalization and absolute quantification in cfMeDIP workflows.	[30]
FSE-EME	Fragment size selection and end-motif enrichment; size-selection/enrichment for mono-nucleosomal-sized fragments or defined end motifs to enrich tumor-derived cfDNA prior to library prep.	[27]
cfDNA NGS panels	Commercial comprehensive cfDNA NGS panels (Guardant360, FoundationOne^®^ Liquid CDx); clinically validated, large, targeted panels for therapy selection, monitoring, and companion diagnostics; include hybrid-capture, UMIs, and bioinformatic QC.	[31,32]
SM-AFD	Single-molecule/amplification-free detection (emerging); experimental approaches that minimize amplification bias (single-molecule sequencing, nanopore/PacBio with specialized prep) for direct detection of fragmented cfDNA.	[27,28]
DNA-PAINT	Super-resolution, single-molecule imaging that uses transient DNA hybridization to detect point mutations, methylation patterns, and amplified sequences at femtomolar sensitivity.	[33,34]

**Table 2 cimb-47-00738-t002:** cftRNA/exosomal RNA detection methods, short description, and representative references (PMID).

Method	Short Description	References
EV-RNA-seq	Exosome/extracellular vesicle (EV) isolation + exoRNA sequencing; enrichment of EVs (ultracentrifugation, size-exclusion, precipitation, commercial kits) followed by RNA extraction and sequencing.	[53]
Targeted RT-qPCR	ddPCR for miRNAs and fusion transcripts; sensitive quantification of defined transcripts using RT-qPCR or ddPCR on exoRNA or total plasma RNA.	[53,54]
UMI-RNA-seq	RNA-Seq with UMIs and low-input library preps; low-input RNA-Seq protocols optimized for extracellular RNA to avoid amplification bias.	[54]
EV-proteomics	Exosomal proteomics for tissue-of-origin determination; mass-spectrometry proteomic profiling of exosome cargo proteins to infer tissue/cell-type origin.	[55,56]
circRNA/lncRNA profiling	Circular RNA and lncRNA in exosomes are resistant to degradation and can yield cancer-specific biomarkers.	[57]
DNA-PAINT-miRNA	DNA-PAINT and DNA-origami nanoarrays for amplification-free detection of miRNA in fluids/exosomes.	[58]
Targeted Fus-RNA-seq	Fusion transcript detection (targeted RNA panels); targeted RNA assays for fusion transcript detection in plasma/exosomes.	[59,60]
Ago-FISH, FRET-FISH	Ago-FISH, FRET-FISH, and other single-molecule RNA detection; single-molecule RNA detection methods for multiplexed, amplification-free detection of short RNA.	[61,62]
exoRNA + proteomics	Integration: multiomic exoRNA + proteomics pipelines; combined workflows analyzing exosomal RNA, DNA, and protein cargo for integrated biomarker signatures.	[63]

**Table 3 cimb-47-00738-t003:** Key Clinical Trials and Longitudinal Studies of cftDNA-based Assays.

Trial/Study	Cancer Type/Population	Assay Type and Methodology	Clinical Objective	Performance Metrics *	Status and Identifier
GRAIL CCGA	Multiple cancer types	cfDNA methylation + sequencing, ML	Early multi-cancer detection and tissue-of-origin prediction	Sensitivity: 39–92% (stage-dependent); specificity: ~99% [107]	NCT02889978
NHS-Galleri	General population, ages 50–77 (UK)	cfDNA methylation classifier	Determine whether cfDNA screening reduces late-stage cancer diagnoses	Interim PPV: ~43%; specificity: ~99.5% [108]	ISRCTN91431511
Natera Signatera (MRD)	Colorectal, breast, lung (post-surgery)	Tumor-informed personalized ctDNA sequencing	Detect MRD, guide adjuvant therapy, predict recurrence	PPV: >95% for recurrence; lead time vs. imaging: ~8–12 months	Multiple ongoing
Guardant ECLIPSE (LUNAR-2/Shield)	Average-risk colorectal cancer screening	cfDNA fragmentomics, methylation, mutation	Early CRC detection for FDA registrational approval	Sensitivity (CRC): ~83%; specificity: ~90%; PPV: ~49% in screening setting [109]	NCT04136002

* Performance metrics vary depending on cancer type, disease stage, and population tested. Values shown are from published interim analyses.

**Table 4 cimb-47-00738-t004:** Retrospective (Pre-diagnosis) Studies Using Banked Plasma for cftDNA Detection.

Cohort (Year)	Design and Timing	Assay/Analyte	Cancer Type(s)	Lead Time Before Dx	N (Cases/Controls)	Key Performance (as reported)	Reference
PanSeer (2020)	Retrospective case–control nested in longitudinal cohort; banked pre-Dx samples	Targeted bisulfite-seq methylation panel (“PanSeer”)	Multi-cancer (stomach, esophageal, colorectal, lung, liver)	Up to 4 years pre-Dx	191/414 (includes pre-Dx positives and non-cancer controls)	Pre-Dx samples from individuals who later developed cancer were frequently positive; high specificity (~95–96%) reported in the controls; sensitivity varies by cancer/stage	[119]
HUNT (2023)	Nested case-control in population study; samples ≤ 24 mo pre-Dx	qMSP methylation markers (e.g., IKZF1, SFRP1/2, VIM, BMP3)	Colorectal cancer	Up to 2 years pre-Dx	106/106 (derivation) + validation set	Panel sensitivity 43%; specificity 86% in derivation; validated in an independent set	[120]
ARIC (2025)	Retrospective analysis of prospectively banked plasma at two timepoints (“Early” ~0–4 mo pre-Dx; “Very Early” 3.1–3.5 y pre-Dx)	Duplex-sequencing mutation panel; WGS-based personalized panels; low-depth WGS aneuploidy	Multi-cancer (colon, rectal, pancreatic, lung, breast, liver)	>3 years in subset; months in all positives	26/26 (matched set)	Early timepoint sensitivity 31% (8/26); 0/26 controls positive; in 4/6 with earlier samples, identical mutations detectable at 8.6–79× lower VAF	[121]

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
