# Peer review of "Multi-Biofluid Approaches for cftDNA and cftRNA Biomarker Detection: Advances in Early Cancer Detection and Monitoring"

_cimb, 2025, doi:10.3390/cimb47090738_

Round 1
Reviewer 1 Report
Comments and Suggestions for Authors
The manuscript by Douglas Ruden comprehensively describes emerging technologies for cell-free tumor DNA and RNA in serum.
- While the title indicates "towards early detection and monitoring", the manuscript fails to adequately describe how the mentioned emerging technologies (especially DNA-PAINT, which is highlighted) can at this stage improve early detection strategies--which are limited by quantitatively low cft DNA and RNA, high FPR.
- While the author mentions spike-in controls in discussion as a major advancement enabling quantification, this technique only prevents assay based misestimation. Isn't it? Therefore, the manuscript would be better if spike-in controls were mentioned as an improvement in existing technology.
- Graphic abstract is poorly placed at the end of results. It does not add much to the manuscript and could be avoided.
- The manuscript could be improved by adding any details on known clinical trials looking at these techniques and if there are any currently ongoing longitudinal studies that would benefit from using these techniques.
- While the title mentions "in serum", the author describes other bodily fluids that could be used to monitor/quantify cft DNA and RNA. Perhaps the title needs to be adjusted to reflect that.
Author Response
CIMB Response to Reviewers 09042025
Reviewer 1:
Comments and Suggestions for Authors
Comment: The manuscript by Douglas Ruden comprehensively describes emerging technologies for cell-free tumor DNA and RNA in serum.
Response: I thank the reviewer for their kind words. Changes in the manuscript are marked with Track Changes or | Left Borders.
Comment: While the title indicates "towards early detection and monitoring", the manuscript fails to adequately describe how the mentioned emerging technologies (especially DNA-PAINT, which is highlighted) can at this stage improve early detection strategies--which are limited by quantitatively low cft DNA and RNA, high FPR.
Response: We thank the reviewer for highlighting the importance of clarifying the current limitations of DNA-PAINT in early detection. We have revised the Discussion to emphasize that DNA-PAINT, while not yet clinically applicable due to low cftDNA abundance and FPR challenges, provides unique advantages such as single-molecule sensitivity and methylation detection that could complement enrichment-based strategies. We also revised the title to reflect that cftDNA and cftRNA can be measured in multiple biofluids beyond serum, aligning the manuscript more closely with its broader scope.
Comment: While the author mentions spike-in controls in discussion as a major advancement enabling quantification, this technique only prevents assay based misestimation. Isn't it? Therefore, the manuscript would be better if spike-in controls were mentioned as an improvement in existing technology.
Response: We appreciate the reviewer’s point and agree that spike-in controls primarily address assay-based misestimation rather than increasing biological signal. We have revised the Discussion to clarify this distinction, emphasizing that spike-ins represent an important improvement in existing technology by enhancing reproducibility and standardization, which are essential for reliable clinical implementation of cftDNA and cftRNA assays.
Comment: Graphic abstract is poorly placed at the end of results. It does not add much to the manuscript and could be avoided.
Response: I removed the Graphic Abstract from the text. I will leave it up to the editor to include it in the final version of the paper. My understanding that it does not show up in the final pdf but rather in the journal to highlight the article. I also moved the two figures to near where they are first mentioned.
Comment: The manuscript could be improved by adding any details on known clinical trials looking at these techniques and if there are any currently ongoing longitudinal studies that would benefit from using these techniques.
Response: : I added a new section, 3.4. Clinical Trials and Longitudinal Studies of cftDNA Detection Methods, and a new Table 4, which both discuss positive predictive values (PPVs). This also addresses a comment by Reviewer 2 who also asked for details of the major cftDNA clinical trials and their PPVs.
Comment: While the title mentions "in serum", the author describes other bodily fluids that could be used to monitor/quantify cft DNA and RNA. Perhaps the title needs to be adjusted to reflect that.
Response: We have revised the title in include all biofluids.
Submission Date
15 August 2025
Date of this review
03 Sep 2025 23:02:01
Reviewer 2:
Comment: Dr. Ruden in this well-written manuscript has described the role of cftDNA/cftRNA in the role of cancer diagnosis and different platforms that can be used to detect tumor specific cftDNA/cftRNA in various bodily fluids.
Response: I appreciate the reviewer’s excellent comments.
Comment: To enhance the manuscript Dr. Ruden should address the suggestions below:
What are the false positive rates/positive values of these assays mentioned? It should be mentioned in the article to give context in the reliability of these assays in predicting cancer diagnosis.
Response: I added a new section, 3.4. Clinical Trials and Longitudinal Studies of cftDNA Detection Methods, and a new Table 4, which both discuss positive predictive values. This also addresses a comment by Reviewer 1 who asked for details of the major cftDNA clinical trials.
Comment: Have there been retrospective studies done on patients who have been diagnosed with cancer where pre-diagnosis samples were tested for cftDNA/cftRNA to see if it could reliably predict diagnosis/outcomes of patients? These types of studies data should be mentioned in the manuscript to show clinical benefit of these assays.
Response: I added a new section, 3.5. Retrospective Pre-diagnosis Studies of cftDNA (from Banked Samples), and a new Table 5, to address this excellent point.
Submission Date
15 August 2025
Date of this review
26 Aug 2025 16:37:27
Reviewer 2 Report
Comments and Suggestions for Authors
Dr. Ruden in this well-written manuscript has described the role of cftDNA/cftRNA in the role of cancer diagnosis and different platforms that can be used to detect tumor specific cftDNA/cftRNA in various bodily fluids. To enhance the manuscript Dr. Ruben should address the suggestions below:
- What are the false positive rates/positive values of these assays mentioned? It should be mentioned in the article to give context in the reliability of these assays in predicting cancer diagnosis.
- Have there been retrospective studies done on patients who have been diagnosed with cancer where pre-diagnosis samples were tested for cftDNA/cftRNA to see if it could reliably predict diagnosis/outcomes of patients? These types of studies data should be mentioned in the manuscript to show clinical benefit of these assays.
Author Response

(The authors gave the same response as above.)
